# Model-based Reinforcement Learning for Semi-Markov Decision Processes with Neural ODEs

**Jianzhun Du,   Joseph Futoma,   Finale Doshi-Velez**
Harvard University
Cambridge, MA 02138
jzdu@g.harvard.edu, {jfutoma, finale}@seas.harvard.edu

## Abstract

We present two elegant solutions for modeling continuous-time dynamics, in a novel model-based reinforcement learning (RL) framework for semi-Markov decision processes (SMDPs) using neural ordinary differential equations (ODEs). Our models accurately characterize continuous-time dynamics and enable us to develop high-performing policies using a small amount of data. We also develop a model-based approach for optimizing time schedules to reduce interaction rates with the environment while maintaining the near-optimal performance, which is not possible for model-free methods. We experimentally demonstrate the efficacy of our methods across various continuous-time domains.

## 1   Introduction

Algorithms for deep reinforcement learning (RL) have led to major advances in many applications ranging from robots [Gu et al., 2017] to Atari games [Mnih et al., 2015]. Most of these algorithms formulate the problem in discrete time and assume observations are available at every step. However, many real-world sequential decision-making problems operate in continuous time. For instance, control problems such as robotic manipulation are generally governed by systems of differential equations. In healthcare, patient observations often consist of irregularly sampled time series: more measurements are taken when patients are sicker or more concerns are suspected, and clinical variables are usually observed on different time-scales.

Unfortunately, the problem of learning and acting in continuous-time environments has largely been passed over by the recent advances of deep RL. Previous methods using semi-Markov decision processes (SMDPs) [Howard, 1964]—including $Q$-learning [Bradtke and Duff, 1995], advantage updating [Baird, 1994], policy gradient [Munos, 2006], actor-critic [Doya, 2000]—extend the standard RL framework to continuous time, but all use relatively simple linear function approximators. Furthermore, as *model-free* methods, they often require large amounts of training data. Thus, rather than attempt to handle continuous time directly, practitioners often resort to discretizing time into evenly spaced intervals and apply standard RL algorithms. However, this heuristic approach loses information about the dynamics if the discretization is too coarse, and results in overly-long time horizons if the discretization is too fine.

In this paper, we take a *model-based* approach to continuous-time RL, modeling the dynamics via neural ordinary differential equations (ODEs) [Chen et al., 2018]. Not only is this more sample-efficient than model-free approaches, but it allows us to efficiently adapt policies learned using one schedule of interactions with the environment for another. Our approach also allows for optimizing the measurement schedules to minimize interaction with the environment while maintaining the near-optimal performance that would be achieved by constant intervention.

Specifically, to build flexible models for continuous-time, model-based RL, we first introduce ways to incorporate *action* and *time* into the neural ODE work of [Chen et al., 2018, Rubanova et al., 2019]. We present two solutions, ODE-RNN (based on a recurrent architecture) and Latent-ODE

(based on an encoder-decoder architecture), both of which are significantly more robust than current approaches for continuous-time dynamics. Because these models include a hidden state, they can handle partially observable environments as well as fully-observed environments. Next, we develop a unified framework that can be used to learn both the *state transition* and the *interval timing* for the associated SMDP. Not only does our model-based approach outperform baselines in several standard tasks, we demonstrate the above capabilities which are not possible with current model-free methods.

## 2 Related work

There has been a large body of work on continuous-time reinforcement learning, many based on the RL framework of SMDPs [Bradtke and Duff, 1995, Parr and Russell, 1998]. Methods with linear function approximators include $Q$-learning [Bradtke and Duff, 1995], advantage updating [Baird, 1994], policy gradient [Munos, 2006], and actor-critic [Doya, 2000]. Classical control techniques such as the linear–quadratic regulator (LQR) [Kwakernaak and Sivan, 1972] also operate in continuous time using a linear model class that is likely too simplistic and restrictive for many real-world scenarios. We use a more flexible model class for learning continuous-time dynamics models to tackle a wider range of settings.

Other works have considered SMDPs in the context of varying time discretizations. Options and hierarchical RL [Sutton et al., 1999, Barto and Mahadevan, 2003] contain temporally extended actions and meta-actions. More recently, Sharma et al. [2017] connected action repetition of deep RL agents with SMDPs, and Tallec et al. [2019] identified the robustness of $Q$-learning on different time discretizations; however, the transition times were still evenly-spaced. In contrast, our work focuses on truly continuous-time environments with irregular, discrete intervention points.

More generally, discrete-time, model-based RL has offered a sample-efficient approach [Kaelbling et al., 1996] for real-world sequential decision-making problems. Recently, RNN variants have become popular black-box methods for summarizing long-term dependencies needed for prediction. RNN-based agents have been used to play video games [Oh et al., 2015, Chiappa et al., 2017]; Ha and Schmidhuber [2018] trained agents in a "dreamland" built using RNNs; Igl et al. [2018] utilized RNNs to characterize belief states in situations with partially observable dynamics; and Neitz et al. [2018] trained a recurrent dynamics model skipping observations adaptively to avoid poor local optima. To our knowledge, no prior work in model-based RL focuses on modeling continuous-time dynamics and planning with irregular observation and action times.

Neural ODEs [Chen et al., 2018] have been used to tackle irregular time series. Rubanova et al. [2019], De Brouwer et al. [2019] used a neural ODE to update the hidden state of recurrent cells; Chen et al. [2018] defined latent variables for observations as the solution to an ODE; Kidger et al. [2020] adjusted trajectories based on subsequent observations with controlled differential equations. To our knowledge, ours is the first to extend the applicability of neural ODEs to RL.

## 3 Background and notation

**Semi-Markov decision processes.** A semi-Markov decision process (SMDP) is a tuple $(\mathcal{S}, \mathcal{A}, \mathcal{P}, \mathcal{R}, \mathcal{T}, \gamma)$, where $\mathcal{S}$ is the state space, $\mathcal{A}$ is the action space, $\mathcal{T}$ is the transition time space and $\gamma \in (0, 1]$ is the discount factor. We assume the environment has transition dynamics $P(s', \tau | s, a) \in \mathcal{P}$ unknown to the agent, where $\tau$ represents the time between taking action $a$ in observed state $s$ and arriving in the next state $s'$ and can take a new action. Thus, we assume no access to any intermediate observations. In general, we are given the reward function $r = R(s, a, s')$ for the reward after observing $s'$. However, in some cases the reward may also depend on $\tau$, i.e. $r = R(s, a, s', \tau)$, for instance if the cost of a system involves how much time has elapsed since the last intervention. The goal throughout is to learn a policy maximizing long-term expected rewards $\mathbb{E}\left[\sum_{i=1}^{L} \gamma^{t_i} r_i\right]$ with a finite horizon $T = t_L$, where $t_i = \sum_{j=1}^{i} \tau_j$.

While the standard SMDP model above assumes full observability, in our models, we will introduce a latent variable $z \in \mathbb{R}^v$ summarizing the history until right before the most recent state, and learn a transition function $\hat{P}(z', \tau | z, a, s)$, treating $s$ as an *emission* of the latent $z$. Introducing this latent $z$ will allow us to consider situations in which: (a) the state $s$ can be compressed, and (b) we only receive partial observations and not the complete state with one coherent model.

**Neural ordinary differential equations.** Neural ODEs define a latent state $z(t)$ as the solution to an ODE initial value problem using a time-invariant neural network $f_\theta$:

$$\frac{dz(t)}{dt} = f_\theta(z(t), t), \quad \text{where } z(t_0) = z_0. \tag{1}$$

Utilizing an off-the-shelf numerical integrator, we can solve the ODE for $z$ at any desired time. In this work, we consider two different neural ODE models as starting points. First, a standard RNN can be transformed to an ODE-RNN [Rubanova et al., 2019]:

$$\tilde{z}_{i-1} = \text{ODESolve}\left(f_\theta, z_{i-1}, \tau_{i-1}\right), \quad z_i = \text{RNNCell}\left(\tilde{z}_{i-1}, s_{i-1}\right), \quad \hat{s}_i = o(z_i). \tag{2}$$

where $\hat{s}$ is the predicted state and $o(\cdot)$ is the emission (decoding) function. An alternate approach, based on an encoder-decoder structure [Sutskever et al., 2014], is the Latent-ODE [Chen et al., 2018]:

$$z_0 \sim q_\phi(z_0|\{s_0\}_{i=0}^L), \quad \{z_i\}_{i=1}^L = \text{ODESolve}\left(f_\theta, z_0, \{\tau_0\}_{i=0}^{L-1}\right), \quad \hat{s}_i = o(z_i), \tag{3}$$

where $q_\phi$ is a RNN encoder and the latent state $z$ is defined by an ODE. The Latent-ODE is trained as a variational autoencoder (VAE) [Kingma and Welling, 2013, Rezende et al., 2014].

The ODE-RNN allows online predictions and is natural for sequential decision-making, though the effect of the ODE network is hard to interpret in the RNN updates. On the other hand, the Latent-ODE explicitly models continuous-time dynamics using an ODE, along with a measure of uncertainty from the posterior over $z$, but the solution to this ODE is determined entirely by the initial latent state.

**Recurrent environment simulator.** None of the above neural ODE models contain explicit actions. We build on the recurrent simulator of Chiappa et al. [2017], which used the following structure for incorporating the effects of actions:

$$z_i = \text{RNNCell}\left(z_{i-1}, a_{i-1}, \dot{s}_i\right), \quad \hat{s}_i = o(z_i), \tag{4}$$

where $\dot{s}$ denotes either the observed state $s$ or the predicted state $\hat{s}$. We can use either $s$ or $\hat{s}$ during training of the recurrent simulator, but only $\hat{s}$ is available for inference. In this work, we generally use the actual observations during training, i.e. $\dot{s} = s$ (this is known as the teacher forcing strategy), but also find that using *scheduled sampling* [Bengio et al., 2015] which switches between choosing the previous observation and the prediction improves performance on some tasks.

## 4 Approach

In this section, we first describe how to construct ODE-based dynamics models for model-based RL that account for both actions and time intervals, overcoming shortcomings of existing neural ODEs. Then, we describe how to use these models for prediction in the original environment, as well as how to transfer to environments with new time schedules and how to optimize environment interaction.

### 4.1 Model definition and learning

We decompose the transition dynamics into two parts, one to predict the time $\tau$ until the next action with corresponding observation, and one to predict the next latent state $z'$:

$$P(z', \tau|z, a, s) = P(z'|z, a, s, \tau) \cdot P(\tau|z, a, s). \tag{5}$$

Using the recurrent environment simulator from Equation 4 we can incorporate an action $a$ into an ODE-RNN or a Latent-ODE to approximate $P(z'|z, a, s, \tau)$ (referred to as the *state transition*). We can also learn and generate transition times $\tau$ using another neural network $g_\kappa(z, a, s)$ based on the current latent state, action and observed state, which approximates $P(\tau|z, a, s)$ (referred to as the *interval timing*). We model them separately yet optimize jointly in a multi-task learning fashion. Specifically, we propose an *action-conditioned* and *time-dependent* ODE-RNN and Latent-ODE for approximating the transition dynamics $P(z', \tau|z, a, s)$.

**ODE-RNN.** Combining Equations 2 and 4, we obtain the following model:

$$\begin{aligned} \tilde{z}_{i-1} &= \text{ODESolve}\left(f_\theta, z_{i-1}, \dot{\tau}_{i-1}\right), \\ z_i &= \text{RNNCell}\left(\tilde{z}_{i-1}, a_{i-1}, \dot{s}_{i-1}\right), \\ \hat{s}_i &= o(z_i), \quad \hat{\tau}_i = g_\kappa(z_i, a_i, \dot{s}_i), \end{aligned} \tag{6}$$

where $\dot{\tau}$ denotes either the observed time interval $\tau$ or the predicted time interval $\hat{\tau}$, similar to $\dot{s}$.

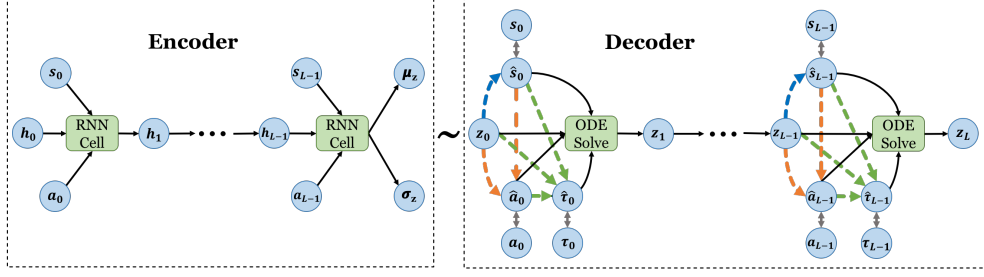

Figure 1: The graphical representation of action-conditioned and time-dependent Latent ODE. Blue dashed arrow represents the emission function. Orange dashed arrow represents the (latent) policy. Green dash arrow represents the prediction of interval timing. Gray double-sided arrow represents the selection between observation and prediction. Note that the encoder is only used in model training.

**Latent-ODE.** Given the parameters $\theta$ of the underlying dynamics, the entire latent trajectory of the vanilla Latent-ODE is determined by the initial condition, $z_0$. However, to be useful as a dynamics model for RL the Latent-ODE should allow for actions to modify the latent state. To do this, at every time we adjust the previous latent state to obtain a new latent state $\tilde{z}$, incorporating the new actions and observations. In particular, we transform the vanilla Latent-ODE using Equations 3 and 4:

$$
\begin{aligned}
z_0 &\sim q_\phi(z_0 | s_0, a_0, s_1, a_1, \ldots, s_{L-1}, a_{L-1}), \\
\tilde{z}_{i-1} &= \xi_\psi(z_{i-1}, a_{i-1}, \dot{s}_{i-1}), \\
z_i &= \text{ODESolve}\left(f_\theta, \tilde{z}_{i-1}, \dot{\tau}_{i-1}\right), \\
\hat{s}_i &= o(z_i), \quad \hat{\tau}_i = g_\kappa(z_i, a_i, \dot{s}_i),
\end{aligned}
\tag{7}
$$

where $\xi_\psi : \mathbb{R}^w \to \mathbb{R}^v$ is a function incorporating the action and observation (or prediction) to the ODE solution. This method of incorporating actions into neural ODEs is similar to *neural controlled differential equations* [Kidger et al., 2020]. We find that a linear transformation works well in practice for $\xi_\psi$, i.e.,

$$
\tilde{z}_{i-1} = W[z_{i-1} \; a_{i-1} \; \dot{s}_{i-1}] + b.
\tag{8}
$$

where $[\cdot]$ is the vector concatenation. The graphical model of Latent-ODE is shown in Figure 1.

The ODE-RNN and Latent ODE intrinsically differ in the main entity they use for modeling continuous-time dynamics. The ODE-RNN models the transition between latent states using a recurrent unit, whereas the Latent-ODE parameterizes the dynamics with an ODE network directly.

**Training objective.** We assume that we have a collection of variable-length trajectories $h = \left(\{s_i^{(n)}\}_{i=0}^{L_n}, \{a_i^{(n)}\}_{i=0}^{L_n-1}, \{\tau_i^{(n)}\}_{i=0}^{L_n-1}\right)_{n=1}^{N}$. We optimize the overall objective in Equation 9 using mini-batch stochastic gradient descent:

$$
\mathcal{L}(h) = \mathcal{L}^{\mathcal{S}}(h) + \lambda \cdot \mathcal{L}^{\mathcal{T}}(h),
\tag{9}
$$

where $\mathcal{L}^{\mathcal{S}}(h)$ is a loss for prediction of state transitions, $\mathcal{L}^{\mathcal{T}}(h)$ is a loss for prediction of interval timings, and $\lambda$ is a hyperparameter trades off between the two. Based on the complexity or importance of predicting state transitions and interval timings, we can emphasize either one by adjusting $\lambda$.

For recurrent models, such as the ODE-RNN, $\mathcal{L}^{\mathcal{S}}(h)$ is simply the mean squared error (MSE); for encoder-decoder models, such as the Latent ODE, it is the negative evidence lower bound (ELBO):

$$
\mathcal{L}^{\mathcal{S}}(h) = \begin{cases}
\frac{1}{N \sum_{n=1}^{N} L_n} \sum_{n=1}^{N} \sum_{i=1}^{L_n} \|\hat{s}_i^{(n)} - s_i^{(n)}\|_2^2 & \text{for recurrent models,} \\
-\mathbb{E}_{z_0^{(n)} \sim q_\phi\left(z_0^{(n)} | \{s_i^{(n)}, a_i^{(n)}\}_{i=0}^{L_n-1}\right)} \left[\log p(\{s_i^{(n)}\}_{i=1}^{L_n} | s_0, \{z_i^{(n)}, a_i^{(n)}, \tau_i^{(n)}\}_{i=0}^{L_n-1})\right] \\
\quad + \mathbb{KL}\left[q_\phi\left(z_0^{(n)} | \{s_i^{(n)}, a_i^{(n)}\}_{i=0}^{L_n-1}\right) \| p(z_0^{(n)})\right] & \text{for encoder-decoder models.}
\end{cases}
\tag{10}
$$

For the loss $\mathcal{L}^{\mathcal{T}}(h)$ for interval timing (Equation 11), we use cross entropy for a small number of discrete $\tau$. Otherwise, MSE can be used for continuous $\tau$.

$$\mathcal{L}^{\mathcal{T}}(h) = \begin{cases} -\frac{1}{N}\sum_{n=1}^{N}\sum_{i=0}^{L_n-1}\sum_{m=1}^{M} y_{i,m}^{(n)} \log p_{i,m}^{(n)} & \text{for classification,} \\ \frac{1}{N\sum_{n=1}^{N}L_n}\sum_{n=1}^{N}\sum_{i=0}^{L_n-1}\|\hat{\tau}_i^{(n)} - \tau_i^{(n)}\|_2^2 & \text{for regression,} \end{cases} \tag{11}$$

where $M$ is the number of classes of time interval, $y_{i,m}^{(n)}$ is the binary indicator if the class label $m$ is the correct classification for $\tau_i^{(n)}$, and $p_{i,m}^{(n)}$ is the predicted probability that $\tau_i^{(n)}$ is of class $m$.

## 4.2 Planning and learning

Now that we have models and procedures for learning them, we move on to the question of identifying an optimal policy. With partial observations, the introduced latent state $z$ provides a representation encoding previously seen information [Ha and Schmidhuber, 2018] and we construct a *latent policy* $\pi_\omega(a|s, z)$ conditioned on $z$; otherwise the environment is fully observable and we construct a policy $\pi_\omega(a|s)$. In general, we model the action-value function $Q(s, a)$ (or $Q([s\ z], a)$) with continuous-time $Q$-learning [Bradtke and Duff, 1995, Sutton et al., 1999] for SMDPs, which works the unequally-sized time gap $\tau$ into the discount factor $\gamma$:

$$Q(s,a) \leftarrow Q(s,a) + \alpha\left[r + \gamma^\tau \max_{a'} Q(s',a') - Q(s,a)\right]. \tag{12}$$

We construct the policy $\pi_\omega$ with a deep $Q$-network (DQN) [Mnih et al., 2015] for discrete actions and deep deterministic policy gradient (DDPG) [Lillicrap et al., 2015] for continuous actions. We perform efficient planning with dynamics models to learn the policy $\pi_\omega$, which is detailed in Section 5. The exact method for planning is orthogonal to the use of our models and framework.

**Transferring to environments with different time schedules.** Our model-based approach allows us to adapt to changes in time interval settings: once we have learned the underlying state transition dynamics $P(z'|z, s, a, \tau)$ from a particular time schedule $p(\tau)$, the model can be used to find a policy for another environment with any different time schedules $p'(\tau)$ (either irregular or regular). The model is generalizable for interval times if and only if it truly learns how the system changes over time in the continuous-time environment.

**Interpolating rewards and optimizing interval times.** In addition to adapting to new time schedules, we can also optimize a state-specific sampling rate to maximize cumulative rewards while minimizing interactions. For example, in Section 5.2, we will demonstrate how we can reduce the number of times a (simulated) HIV patient must come into the clinic for measurement and treatment adjustment while still staying healthy. However, this approach may not work well in situations where constant adjustments on small time scales can hurt performance (e.g., Atari games use frameskipping to avoid flashy animations).

When optimizing interval times, we assume that $\tau$ are discrete, $\tau \in \{1, 2, \ldots, \delta\}$, and we only have access to the immediate reward function $r = R(s, a, s')$. We can optimize the interval times to decrease the amount of interaction with the environment while achieving near-optimal performance, obtained by maximal interaction ($\tau \equiv 1$). Specifically, *assuming we always take an action $a$ for each of the $\tau$ steps in the interval starting from the state $s_0$*, we select the optimal $\tau^*$ based on the estimated $\tau$-step ahead value:

$$\tau^* = \underset{\tau \in [1,2,\ldots,\delta]}{\arg\max} \sum_{i=1}^{\tau} \gamma^{i-1}\hat{r}_i + \gamma^\tau Q^\pi(\hat{s}_\tau, \hat{a}_\tau), \quad \text{where } \hat{r}_i = R(\hat{s}_{i-1}, a, \hat{s}_i), \tag{13}$$

$\hat{s}$ is the simulated state from the model ($\hat{s}_0 = s_0$), $\hat{a}$ is the optimal action at state $\hat{s}$, and $Q^\pi$ is the action-value function from the policy. We interpolate intermediate rewards using the dynamics model—we can simulate by varying $\tau$ what states would be passed through, and what the cumulative reward would be—whereas it is not possible to do this for model-free methods as we have no access to in-between observations. In this way, agents learn to skip situations in which no change of action is needed and minimize environment interventions. The full procedure can be found in Appendix A.1. Note that Equation 13 can be easily extended to continuous interval times if they are lower-bounded, i.e., $\tau \in [a, \infty)$ ($a \in \mathbb{R}^+$); we leave this to future work.

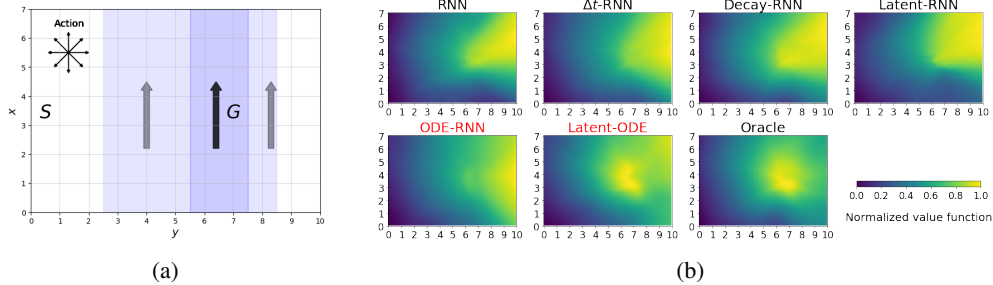

(a)                                                                    (b)

Figure 2: (a) 2D continuous windy gridworld, where "S" and "G" is the start area ($x \in [3, 4], y \in [0, 1]$) and goal area ($x \in [3, 4], y \in [6.5, 7.5]$), and the shaded region and arrows represent the wind moves upward (the darker color indicates the stronger wind); (b) normalized value functions obtained by DQNs for all baselines and our proposed ODE-based models (marked as red) over the gridworld. "Oracle" refers to the model-free baseline trained until convergence. The lighter the pixel, the higher the value. The policy developed with the Latent-ODE is "closet" to the optimal policy (oracle).

## 5 Experiments

We evaluate our ODE-based models across four continuous-time domains. We show our models characterize continuous-time dynamics more accurately and allow us to find a good policy with less data. We also demonstrate capabilities of our model-based methods that are not possible for model-free methods.

### 5.1 Experimental setup

**Domains.** We provide demonstrations on three simpler domains—windy gridworld [Sutton and Barto, 2018], acrobot [Sutton, 1996], and HIV [Adams et al., 2004]—and three Mujoco [Todorov et al., 2012] locomotion tasks—Swimmer, Hopper and HalfCheetah—interfaced through OpenAI Gym [Brockman et al., 2016]. Unless otherwise stated, the state space of all tasks is fully observable, and we are given the immediate reward function $r = R(s, a, s')$. We provide more details of domains in Appendix B.

- *Windy gridworld* (Figure 2a). We consider a continuous state version in which agents pursue actions $a = (\Delta x, \Delta y)$ for $\tau \sim \text{Unif}\{1, 7\}$ seconds to reach a goal region despite a crosswind.

- *Acrobot*. Acrobot aims to swing up a pendulum to reach a given height. The dynamics is defined by a set of first-order ODEs and the original version uses $\tau = 0.2$ to discretize underlying ODEs. We instead sample the time interval $\tau$ randomly from $\{0.2, 0.4, 0.6, 0.8, 1\}$.

- *HIV*. Establishing effective treatment strategies for HIV-infected patients based on markers from blood tests can be cast as an RL problem [Ernst et al., 2006, Parbhoo et al., 2017, Killian et al., 2017]. The effective period $\tau$ varies from one day to two weeks. Healthy patients with less severe disease status may only need occasional inspection, whereas unhealthy patients require more frequent monitoring and intervention.

- *Mujoco*. We use action repetition [Sharma et al., 2017] to introduce irregular transition times to Mujoco tasks, however, we assume the intermediate observations are not available so that the dynamics naturally fits into the SMDP definition. The same action repeats $\tau = h(\|\theta_v\|_2)$ times where $\theta_v$ is the angle velocity vector of all joints and $h$ is a periodic function. The periodicity is learnable by RNNs [Gers et al., 2002] and ensures consistently irregular measurements in the course of policy learning[1].

As proofs of concept, we assume the transition times in gridworld and acrobot problems are known, and only focus on learning the state transition probability $P(z'|z, a, s, \tau)$ (set $\lambda = 0$ in Equation 9); for HIV and Mujoco tasks, we learn both state transitions and interval timings.

**Baselines.** We compare the performance of our proposed ODE-based models with four baselines embedded in our model-based RL framework for SMDPs. With the recurrent architecture, we

Table 1: The state prediction error (mean $\pm$ std, over five runs) of all models on three simpler domains. Note that models always consume predictions $\hat{s}$ in testing to match the inference procedure. The Latent-ODE achieves the lowest prediction errors on Acrobot and HIV tasks.

| | RNN | $\Delta t$-RNN | Decay-RNN | Latent-RNN | ODE-RNN | Latent-ODE |
|---|---|---|---|---|---|---|
| Gridworld | $0.894 \pm 0.023$ | $\mathbf{0.334 \pm 0.023}$ | $0.899 \pm 0.022$ | $1.161 \pm 0.039$ | $0.452 \pm 0.040$ | $0.845 \pm 0.017$ |
| Acrobot | $0.176 \pm 0.010$ | $0.039 \pm 0.006$ | $0.060 \pm 0.006$ | $0.176 \pm 0.010$ | $\mathbf{0.022 \pm 0.005}$ | $\mathbf{0.021 \pm 0.005}$ |
| HIV | $0.332 \pm 0.013$ | $0.168 \pm 0.014$ | $0.346 \pm 0.022$ | $0.361 \pm 0.017$ | $0.068 \pm 0.006$ | $\mathbf{0.020 \pm 0.001}$ |

Table 2: The cumulative reward (mean $\pm$ std, over five runs) of policies developed with all models on three domains. "Oracle" (*italic*) refers to the model-free baseline trained until convergence. (a) planning in the original irregular time schedule; (b) planning using pretrained models from the original irregular time schedule for a new regular time schedule (gridworld: $\tau = 7$; acrobot: $\tau = 0.2$; HIV: $\tau = 5$).

| | | RNN | $\Delta t$-RNN | Decay-RNN | Latent-RNN | ODE-RNN | Latent-ODE | Oracle |
|---|---|---|---|---|---|---|---|---|
| (a) | Gridworld | $-54.02 \pm 9.24$ | $-45.64 \pm 8.22$ | $-48.91 \pm 8.97$ | $-47.92 \pm 8.16$ | $-58.50 \pm 10.46$ | $\mathbf{-34.87 \pm 1.96}$ | $-34.17 \pm 1.47$ |
| | Acrobot | $-179.35 \pm 10.49$ | $-106.78 \pm 10.34$ | $-105.23 \pm 10.96$ | $-181.64 \pm 10.92$ | $-65.90 \pm 7.06$ | $\mathbf{-54.26 \pm 4.01}$ | $-48.67 \pm 3.29$ |
| | HIV ($\times 10^7$) | $0.78 \pm 0.04$ | $0.75 \pm 0.17$ | $0.95 \pm 0.21$ | $0.82 \pm 0.05$ | $11.74 \pm 1.50$ | $\mathbf{30.32 \pm 2.70}$ | $35.22 \pm 1.42$ |

| | | RNN | $\Delta t$-RNN | Decay-RNN | Latent-RNN | ODE-RNN | Latent-ODE | Oracle |
|---|---|---|---|---|---|---|---|---|
| (b) | Gridworld | $-61.01 \pm 10.03$ | $-64.55 \pm 10.89$ | $-60.78 \pm 10.03$ | $-52.32 \pm 8.91$ | $-114.70 \pm 11.65$ | $\mathbf{-49.31 \pm 6.62}$ | $-35.93 \pm 1.95$ |
| | Acrobot | $-407.46 \pm 13.82$ | $-281.92 \pm 9.99$ | $-285.07 \pm 8.47$ | $-237.25 \pm 10.29$ | $-190.82 \pm 9.13$ | $\mathbf{-171.37 \pm 10.07}$ | $-78.75 \pm 3.23$ |
| | HIV ($\times 10^7$) | $7.66 \pm 1.79$ | $17.21 \pm 2.44$ | $5.84 \pm 1.62$ | $16.95 \pm 3.05$ | $11.32 \pm 1.09$ | $\mathbf{21.60 \pm 2.39}$ | $33.55 \pm 1.97$ |

evaluate 1) vanilla RNN; 2) $\Delta t$-RNN, where the time intervals are concatenated with the original input as an extra feature; 3) Decay-RNN [Che et al., 2018], which adds an exponential decay layer between hidden states: $\tilde{z} = e^{-\max\{0, w\tau + b\}} \odot z$. With the encoder-decoder architecture, we evaluate 4) Latent-RNN, where the decoder is constructed with a RNN and the model is trained variationally. RNNs in all models are implemented by gate recurrent units (GRUs) [Cho et al., 2014]. Moreover, we also run a model-free method (DQN or DDPG) for comparison.

## 5.2 Demonstrations on simpler domains

We learn the *world model* [Ha and Schmidhuber, 2018] of simpler environments for planning. We gather data from an initial random policy and learn the dynamics model on this collection of data. Agent policies are *only* trained using fictional samples generated by the model *without considering the planning horizon*. To achieve optimal performance (i.e. the model-free baseline trained until convergence), the model has to capture long-term dependencies so that the created virtual world is accurate enough to mimic the actual environment. Thus, with this planning scheme, we can clearly demonstrate a learned model's capacity. The details of the algorithm can be found in Appendix A.2.

**Latent-ODEs mimic continuous-time environments and value functions more accurately.** Because the training dataset is fixed, we can use the *state prediction error* (MSE in Equation 10) on a separate test dataset to measure if the model learns the dynamics well. Table 1 shows state prediction errors of all models. The ODE-RNN and Latent-ODE outperform other models on acrobot and HIV, but the $\Delta t$-RNN achieves the lowest error on the windy gridworld. However, by visualizing the value functions of learned policies which are constructed using dynamics models (Figure 2b), we find that only the Latent-ODE accurately recovers the true gridworld (the one from the model-free baseline), whereas the $\Delta t$-RNN characterizes the dark parts of the world very well, but fails to identify the true goal region (the light part). Thus, the lower state prediction error averaged over the gridworld does not imply better policies.

**Latent-ODEs help agents develop better policies.** Table 2a shows the performance of all models, in terms of returns achieved by policies in the actual environment. Latent-ODE consistently surpasses other models and achieves near-optimal performance with the model-free baseline. In contrast, all non-ODE-based models develop very poor policies on the acrobot and HIV problems.

**Latent-ODEs are more robust to changes in time intervals.** To test if the dynamics model is generalizable across interval times, we change the time schedules from irregular measurements to

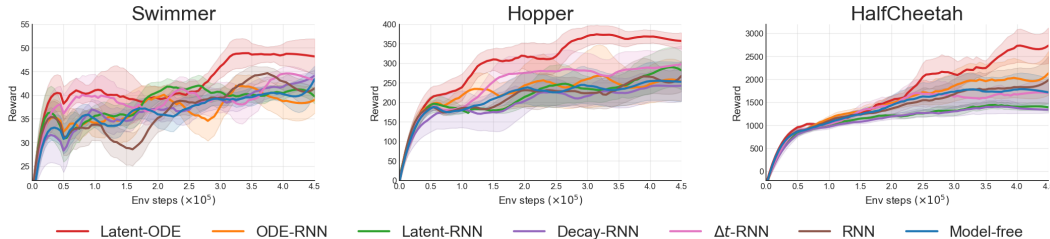

Figure 4: Learning curves with all models on three Mujoco tasks. The shaded region represents a standard deviation of average evaluation over four runs (evaluation data is collected every 5000 timesteps). Curves are smoothed with a 20-point window for visual clarity. The Latent-ODE develop better policies with fewer data than the model-free method and other models on all three tasks.

regular measurements without retraining the models. Due to the page limit, we show the results of $\tau = 7$ for the gridworld, $\tau = 0.2$ for the acrobot and $\tau = 5$ for HIV in Table 2b and include full results of all time discretizations in Appendix D.2. The Latent-ODE is once again the best dynamics model to solve the new environment, even if the transition times are regular.

**Optimized time schedules achieve the best balance of high performance and low interaction rate.** For evaluation, to ensure the fair comparison of different interaction rates given the fixed horizon, we collect the reward at every time step (every day) from the environment and calculate the overall cumulative reward. The results on the HIV environment are shown in Figure 3. Developing the model-based schedule using the Latent-ODE, we can obtain similar returns as measuring the health state every two days, but with less than half the interventions. Further, using the oracle dynamics, the optimized schedule achieves similar performance with maximal interaction ($\tau = 1$) while reducing interaction times by three-quarters.

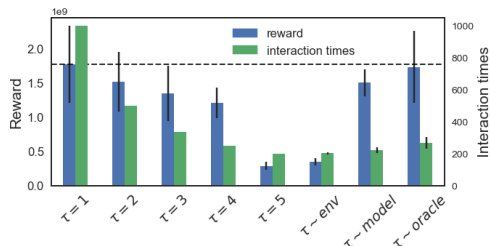

Figure 3: The cumulative reward vs. interaction rate on the HIV environment. "$\tau \sim$ env" refers to the original time schedule for training models; "$\tau \sim$ model/oracle" refers to selecting $\tau$ using the Latent-ODE/true dynamics and Equation 13.

**Latent variables capture hidden state representations in partially observable environment.** We mask two blood test markers in the state space to build a partially observable HIV environment, where we demonstrate the behavior of the latent policy $\pi^{MB}(a|s_{\text{partial}}, z)$. We compare its performance with the model-based policy $\pi^{MB}(a|s_{\text{partial}})$ and the model-free policy $\pi^{MF}(a|s_{\text{partial}})$ using partial observations, and the model-based policy $\pi^{MB}(a|s_{\text{full}})$ using full observations. All model-based policies are developed with the Latent-ODE. Based on the results in Table 3, the latent variable $z$ improves the performance in the

Table 3: The performance of different policies on partially observable HIV task.

| Policy | Reward ($\times 10^7$) |
|---|---|
| $\pi^{MB}(a\|s_{\text{partial}})$ | 12.88 ± 1.71 |
| $\pi^{MB}(a\|s_{\text{partial}}, z)$ | **18.04 ± 2.07** |
| $\pi^{MF}(a\|s_{\text{partial}})$ | *16.96 ± 1.53* |
| $\pi^{MB}(a\|s_{\text{full}})$ | *30.32 ± 2.70* |

partially observable setting, and even builds a better policy than the model-free baseline, though the latent policy cannot achieve the asymptotic performance of the policy using full observations.

## 5.3 Continuous control: Mujoco

For the more complex Mujoco tasks, exploration and learning must be interleaved. We combine model predictive control (MPC) [Mayne et al., 2000] with the actor-critic method (DDPG) for planning. MPC refines the stochastic model-free policy (the actor) by sampling-based planning [Wang and Ba, 2019, Hong et al., 2019], and the value function (the critic) mitigates the short sight of imaginary model rollouts in MPC planning [Lowrey et al., 2018, Clavera et al., 2020]. This approach iterates between data collection, model training, and policy optimization, which allows us to learn a good policy with a small amount of data. The details of the algorithm can be found in Appendix A.3.

**Latent-ODEs exhibit the sample efficiency across all three Mujoco tasks.** Figure 4 shows the learning process of all models on Mujoco tasks. The Latent-ODE is more sample-efficient than the model-free method and other models on all three tasks. For example, on the swimmer and hopper task, we develop a high-performing policy over 100k environment steps using the Latent-ODE, whereas the model-free baseline requires four times the amount of data. However, the ODE-RNN is not as good as the Latent-ODE and its performance is similar with other baselines.

## 6  Discussion and conclusion

We incorporate actions and time intervals into neural ODEs for modeling continuous-time dynamics, and build a unified framework to train dynamics models for SMDPs. Our empirical evaluation across various continuous-time domains demonstrates the superiority of the Latent-ODE in terms of model learning, planning, and robustness for changes in interval times. Moreover, we propose a method to minimize interventions with the environment but maintain near-optimal performance using the dynamics model, and show its effectiveness in the health-related domain.

The flexibility of our model training procedure (Section 4.1), which is orthogonal to the planning method, inspires the future research on model-based RL and neural ODEs in various aspects. First, one might easily enhance the performance of Latent-ODEs in continuous-time domains using more advanced planning schemes and controllers, e.g., TD3 [Fujimoto et al., 2018], soft actor-critic [Haarnoja et al., 2018] and etc. Second, since our model is always trained on a batch of transition data, we can apply our method to the continuous-time off-policy setting with the recent advances on model-based offline RL [Yu et al., 2020, Kidambi et al., 2020]. RL health applications (e.g., in ICU [Gottesman et al., 2020]) might benefit from this in particular, because practitioners usually assume the presence of a large batch of already-collected data, which consists of patient measurements with irregular observations and actions. Furthermore, while we focus on flat actions in this work, it is natural to extend our models and framework to model-based hierarchical RL [Botvinick and Weinstein, 2014, Li et al., 2017].

Last but not least, we find that training a Latent-ODE usually takes more than ten times longer than training a simple RNN model due to slow ODE solvers, which means the scalability might be a key limitation for applying our models to a larger state space setting. We believe that the efficiency of our methods will not only be significantly improved with a theoretically faster numerical ODE solver (e.g., [Finlay et al., 2020, Kelly et al., 2020]), but also with a better implementation of ODE solvers[2] (e.g., a faster C++/Cython implementation, using single precision arithmetic for solvers, and etc.).

## Broader Impact

We introduce a new approach for continuous-time reinforcement learning that could eventually be useful for a variety of applications with irregular time-series, e.g. in healthcare. However, models are only as good as the assumptions made in the architecture, the data they are trained on, and how they are integrated into a broader context. Practitioners should treat any output from RL models objectively and carefully, as in real life there are many novel situations that may not be covered by the RL algorithm.

## Acknowledgement

We thank Andrew Ross, Weiwei Pan, Melanie Pradier and other members from Harvard Data to Actionable Knowledge lab for helpful discussion and feedbacks. We thank Harvard Faculty of Arts and Sciences Research Computing and School of Engineering and Applied Sciences for providing computational resources. FDV and JD are supported by an NSF CAREER.

## Footnotes

[1]Otherwise $\tau$ remains unchanged after robots learn to run, because $\|\theta_v\|_2$ only fluctuates in a small range.

[2]We use the implementation of ODE solvers from Python torchdiffeq library.

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
