[Supplementary Material]

# Supplementary Materials

## A  Algorithm details

### A.1  Optimizing interval times

---

**Algorithm 1:** Optimizing interval times with the dynamics model.

---

**Input** : The pretrained state transition dynamics model $\hat{P}(z'|z, a, s, \tau)$, the replay buffer $\mathcal{D}$, the
reward function $R(s, a, s')$, the time horizon $T$, and the number of episodes $N_e$.
**Output** : $\pi_\omega(a|s)$.

1 Initialize the policy $\pi_\omega(a|s)$ (along with the action-value function $Q^\pi(s, a)$);
2 **for** $j \leftarrow 1$ **to** $N_e$ **do**
3      $t \leftarrow 0$;
4      Observe the initial state $s$ from the environment;
5      Initialize the latent state $z$;
6      **while** $t < T$ **do**
7          Select action $a \sim \pi_\omega(\cdot|s)$;
8          $\hat{s}_0 \leftarrow s$;
9          **for** $i \leftarrow 1$ **to** $|\mathcal{T}|$ **do**
10              $z_i \leftarrow \hat{P}(\cdot|z, a, s, i)$;
11              Decode $\hat{s}_i$ from $z_i$;
12              Select action $\hat{a}_i \sim \pi_\omega(\cdot|\hat{s}_i)$;
13              Calculate the immediate reward $\hat{r}_i \leftarrow R(\hat{s}_{i-1}, a, \hat{s}_i)$ at the current time point $t + i$;
14          **end**
15          Select the best incoming time interval $\tau^* \leftarrow \arg\max_\tau \sum_{i=1}^{\tau} \gamma^{i-1}\hat{r}_i + \gamma^\tau Q^\pi(\hat{s}_\tau, \hat{a}_\tau)$;
16          $z', s', r \leftarrow z_{\tau^*}, \dot{s}_{\tau^*}, \sum_{i=1}^{\tau^*} \gamma^{i-1}\dot{r}_i$;
17          **if** $s'$ *is not the terminated state* **then**
18              Store the tuple $(s, a, s', r, \tau)$ into $\mathcal{D}$;
19          **else**
20              Store the tuple $(s, a, \text{NULL}, r, \tau)$ into $\mathcal{D}$;
21              **break**;
22          **end**
23          Optimize $\pi_\omega$ with data in $\mathcal{D}$;
24          $t, s, z \leftarrow t + \tau^*, s', z'$;
25      **end**
26 **end**

---

Our innovation of optimizing interval times is highlighted in blue in Algorithm 1. Note that we can use either the imaginary reward $\hat{r}$ from the dynamics model or the true reward $r$ from the environment for training the policy $\pi_\omega$, i.e., $\dot{r}$ can be either $\hat{r}$ or $r$, and similar case for $\dot{s}_{\tau^*}$ (line 16 of Algorithm 1). In this work, to focus on the efficacy of the optimized time schedules, we use the true reward $r$ and true observation $s_{\tau^*}$; but the optimal $\tau$ is always determined using the imaginary reward $\hat{r}$.

A key assumption of Algorithm 1 is that acting often using short time intervals will not hurt performance, and that maximal interaction (i.e. $\tau = 1$ for discrete time) will have the optimal performance. In many scenarios, this assumption seems reasonable and applying Algorithm 1 may work well. For instance, in healthcare applications, it is safe to assume that more frequent monitoring of patients and more careful tuning of their treatment plan should yield optimal performance, although in practice this is not generally done due to e.g. resource constraints. However, this assumption does not hold for all environments. For example, some Atari games require frameskipping, i.e., repeating actions for $k$ times, because we need enough changes in pixels to find a good policy. Also, some control problems have an optimal time step for the underlying physical systems. In these situations, one might set a minimum threshold $a$ for $\tau$ and apply the algorithm, i.e., $\tau \in [a, +\infty)$; together with the extension to continuous time (Equation 14), we leave them to future work.

$$\tau^* := \underset{\tau \in [a, \infty)}{\arg\max} \int_{t=a}^{\tau} \gamma^t \hat{r} \, dt + \gamma^\tau Q^\pi(\hat{s}_\tau, \hat{a}_\tau) \tag{14}$$

## A.2 Learning world models

---

**Algorithm 2:** Learning world models [Ha and Schmidhuber, 2018] for SMDPs.

---
**Input** : The replay buffer $\mathcal{D}$, the reward function $R(s, a, s')$, the time horizon $T$, and the number of episodes $N_t$ for model learning and $N_e$ for policy optimization.

**Output** : $\hat{P}(z', \tau | z, a, s), \pi_\omega(a|s)$.

1 Initialize the policy $\pi_\omega(a|s)$ (along with the action-value function $Q^\pi(s, a)$);
2 Initialize the dynamics model $\hat{P}(z', \tau | z, a, s)$;
3 Collect a collection of trajectories $h = \left( \{s_i^{(n)}\}_{i=0}^{L_n}, \{a_i^{(n)}\}_{i=0}^{L_n-1}, \{\tau_i^{(n)}\}_{i=0}^{L_n-1} \right)_{n=1}^{N_t}$ using
   random policies, where $\sum_{i=0}^{L_n-1} \tau_i^{(n)} < T$ for $\forall n \in \{1, 2, \dots, N_t\}$;
4 Train $\hat{P}(z', \tau | z, a, s)$ with $h$ as described in Section 4.1;
5 **for** $i \leftarrow 1$ **to** $N_e$ **do**
6      $t \leftarrow 0$;
7      Observe the initial state $s$ from the environment or sample from a set of initial states;
8      Initialize the latent state $z$;
9      **while** $t < T$ **do**
10          Select action $a \sim \pi_\omega(\cdot|s)$;
11          Predict the incoming time interval $\tau$ and next latent state $z'$ using $\hat{P}(\cdot|z, a, s)$;
12          Decode $s'$ from $z'$;
13          Calculate the reward $r \leftarrow R(s, a, s')$;
14          **if** $s'$ *is not the terminated state* **then**
15             Store the tuple $(s, a, s', r, \tau)$ into $\mathcal{D}$;
16          **else**
17             Store the tuple $(s, a, \text{NULL}, r, \tau)$ into $\mathcal{D}$;
18             **break**;
19          **end**
20          Optimize $\pi_\omega$ with data in $\mathcal{D}$;
21          $t, s, z \leftarrow t + \tau, s', z'$;
22      **end**
23 **end**

---

Algorithm 2 assumes that the dynamics can be fully covered by random policies. However, these may be far away from the optimal policy. Because the policy is trained only on fictional samples without considering the planning horizon, there is no difference between learning in the virtual world created by the dynamics model and the true environment. Thus, the performance of learned policies is mainly determined by the model's capacity.

Nevertheless, Algorithm 2 does not work well on more sophisticated Mujoco tasks, because the dynamics cannot be fully explored by a random policy, and a long planning horizon makes the compounding error of fictional samples accumulate very quickly. One might use an iterative training procedure of Algorithm 2 [Ha and Schmidhuber, 2018, Schmidhuber, 2015] for more complex environments, which interleaves exploration and learning. However, in order to combat the model bias, this type of Dyna-style algorithm usually requires computationally expensive model ensembles [Kurutach et al., 2018, Janner et al., 2019]. Thus, we turn to the MPC-style algorithm for planning (Algorithm 3), which is also sufficiently effective to demonstrate a learned model's capacity and is more computationally efficient.

### A.3 Model predictive control with actor-critic

---

**Algorithm 3:** Model predictive control (MPC) with DDPG for SMDPs.

---

**Input** : The replay buffer $\mathcal{D}$, the environment dataset $\mathcal{D}_{env}$, the reward function $R(s, a, s')$, the planning horizon $H$, the search population $K$, the number of environment steps $M$ and the number of epochs $E$.

**Output** : $\hat{P}(z', \tau|z, a, s), \pi_\omega(a|s), Q_\varphi^\pi(s, a)$.

1  Initialize the actor $\pi_\omega(a|s)$ and the critic $Q_\varphi^\pi(s, a)$, and their target networks $\pi_{\omega'}$ and $Q_{\varphi'}^\pi$;

2  Initialize the dynamics model $\hat{P}(z', \tau|z, a, s)$;

3  Gather a collection of trajectories using random policies, and save them into $\mathcal{D}_{env}$;

4  **for** $i \leftarrow 1$ **to** $E$ **do**

5      Train $\hat{P}(z', \tau|z, a, s)$ with data in $\mathcal{D}_{env}$ as described in Section 4.1;

6      Observe the initial state $s$ from the environment;

7      Initialize the latent state $z$;

8      **for** $j \leftarrow 1$ **to** $M$ **do**

9         **for** $k \leftarrow 1$ **to** $K$ **do**

10           $t_0^{(k)}, \hat{s}_0^{(k)}, z_0^{(k)} \leftarrow 0, s, z$;

11           **for** $h \leftarrow 1$ **to** $H$ **do**

12              Select action $a_{h-1}^{(k)} \sim \pi_\omega(\cdot|\hat{s}_{h-1}^{(k)})$;

13              $z_h^{(k)}, \tau_{h-1}^{(k)} \leftarrow \hat{P}(\cdot|z_{h-1}^{(k)}, a_{h-1}^{(k)}, \hat{s}_{h-1}^{(k)})$;

14              Decode $\hat{s}_h^{(k)}$ from $z_h^{(k)}$;

15              Calculate the reward $\hat{r}_{h-1}^{(k)} \leftarrow R(\hat{s}_{h-1}^{(k)}, a_{h-1}^{(k)}, \hat{s}_h^{(k)})$;

16              $t_h^{(k)} \leftarrow t_{h-1}^{(k)} + \tau_{h-1}^{(k)}$;

17           **end**

18           Select action $a_H^{(k)} \leftarrow \pi_\omega(\cdot|\hat{s}_H^{(k)})$;

19        **end**

20        Select the best sequence index $k^* \leftarrow \arg\max_k \sum_{h=1}^H \gamma^{t_{h-1}^{(k)}} \hat{r}_{h-1}^{(k)} + \gamma^{t_H^{(k)}} Q_\varphi^\pi(\hat{s}_H^{(k)}, a_H^{(k)})$;

21        Select the best action $a \leftarrow a_0^{(k^*)}$;

22        Observe the incoming time interval $\tau$ and next observation $s'$;

23        Encode the next latent state $z' \leftarrow \hat{P}(\cdot|z, a, s, \tau)$;

24        Calculate the reward $r \leftarrow R(s, a, s')$;

25        **if** $s'$ *is not the terminated state* **then**

26           Store the tuple $(s, a, s', r, \tau)$ into $\mathcal{D}$;

27        **else**

28           Store the tuple $(s, a, \text{NULL}, r, \tau)$ into $\mathcal{D}$;

29           Observe the initial state $s$ from the environment;

30           Initialize the latent state $z$;

31           **continue**;

32        **end**

33        Optimize $\pi_\omega$ and $Q_\varphi^\pi$ with data in $\mathcal{D}$;

34        Update target networks $\pi_{\omega'}$ and $Q_{\varphi'}^\pi$;

35        $s, z \leftarrow s', z'$;

36     **end**

37     Store trajectories collected in the current epoch into $\mathcal{D}_{env}$;

38 **end**

---

In Algorithm 3, the actor provides a good initialization of deterministic action sequences, and MPC searches for the best one from these sequences plus Gaussian noises, i.e., we sample the action from a normal distribution $a \sim \mathcal{N}(\pi_\omega(\cdot|s), \sigma^2)$, where $\sigma^2$ is the noise variance (line 12 of Algorithm 3). Further, in the vanilla MPC, the best sequence is determined by the cumulative reward of model rollouts, however, the selected action may not be globally optimal due to the short planning horizon $H$. The critic offers an estimate of expected returns for the final state of simulated trajectories,

which overcomes the shortsighted planning problem (line 20 of Algorithm 3). Note that we use the deterministic action $a_H$ (without Gaussian noise) for the final state $\hat{s}_H$ to calculate the action-value function $Q(\hat{s}_H, a_H)$ (line 18 of Algorithm 3).

## B   Environment specifications

**Windy gridworld.**   We extend the $7 \times 10$ gridworld to have continuous states $s = (x, y)$ and continuous-time actions. That is, picking an orientation (action) $a = (\Delta x, \Delta y)$, agents move towards this direction over arbitrary second(s) $\tau \in \text{Unif}\{1, 7\}$ for $(\tau \Delta x, \tau \Delta y)$. The agent is allowed to "move as a king", i.e., take eight actions, including moving up, down, left, right, upleft, upright, downleft and downright. Specifically, the agent can move $a = (\Delta x, \Delta y)$ in the gridworld every second:

$$a = (\Delta x, \Delta y) = \begin{cases} (0.17, 0) & \text{up}, \\ (-0.17, 0) & \text{down}, \\ (0, -0.17) & \text{left}, \\ (0, 0.17) & \text{right}, \\ (\frac{0.17}{\sqrt{2}}, -\frac{0.17}{\sqrt{2}}) & \text{upleft}, \\ (\frac{0.17}{\sqrt{2}}, \frac{0.17}{\sqrt{2}}) & \text{upright}, \\ (-\frac{0.17}{\sqrt{2}}, -\frac{0.17}{\sqrt{2}}) & \text{downleft}, \\ (-\frac{0.17}{\sqrt{2}}, \frac{0.17}{\sqrt{2}}) & \text{downright}. \end{cases}$$

Every second, agents in the region with the weaker wind ($x \in [2.5, 8.5], y \in [0, 7]$) are pushed to move upward for $(\Delta x, \Delta y) = (\frac{0.17}{4}, 0)$, and agents in the region with the strong wind ($x \in [5.5, 7.5], y \in [0, 7]$) are pushed to move upward for $(\Delta x, \Delta y) = (\frac{0.17}{2}, 0)$. If agents hit the boundary of the gridworld, they will just stand still until the end of transition time $\tau$. Every second in the gridworld incurs -1 cost until discovering the goal region (trigger +10 reward) or after $T = 150$ seconds. Thus, we are given the reward function

$$R(s, a, s', \tau) = \begin{cases} 10 - \tau, & s' = (x \in [3, 4], y \in [6.5, 7.5]), \\ -\tau, & \text{otherwise.} \end{cases}$$

In addition, we feed the zero-centered state $(\bar{x}, \bar{y})$ for both model training and policy optimization.

**Acrobot.**   Acrobot, a canonical RL and control problem, is a two-link pendulum with only the second joint actuated. Initially, both links point downwards. The goal is to swing up the pendulum by applying a positive, neutral, or negative torque on the joint such that the tip of the pendulum reaches a given height. The state space consists of four continuous variables, $s = (\theta_1, \theta_2, \dot{\theta}_1, \dot{\theta}_2)$, where $\theta_1 \in [-\pi, \pi]$ is the angular position of the first link in relation to the joint, and $\theta_2 \in [-\pi, \pi]$ is the angular position of the second link in relation to the first; $\dot{\theta}_1 \in [-4\pi, 4\pi]$ and $\dot{\theta}_2 \in [-9\pi, 9\pi]$ are the angular velocities of each link respectively. The reward is collected as the height of the tip of the pendulum (as recommended in the work of [Wang et al., 2019]) after the transition time $\tau$:

$$R(s, a, s') = -\cos \theta_1 - \cos(\theta_1 + \theta_2),$$

until the goal is reached or after $T = 100$.

**HIV.**   The interaction of the immune system with the human immunodeficiency virus (HIV) and treatment protocols was mathematically formulated as a dynamical system [Adams et al., 2004], which can be resolved using RL approaches [Ernst et al., 2006, Parbhoo et al., 2017, Killian et al., 2017]. The goal of this task is to determine effective treatment strategies for HIV infected patients based on critical markers from a blood test, including the viral load ($V$, which is the main maker indicating if healthy), the number of healthy and infected CD4+ T-lymphocytes ($T_1, T_1^*$, respectively), the number of healthy and infected macrophages ($T_2, T_2^*$, respectively), and the number of HIV-specific cytotoxic T-cells ($E$), i.e., $s = (T_1, T_2, T_1^*, T_2^*, V, E)$. To build a partially observerable HIV environment, $T_1^*$ and $T_2^*$ are removed from the state space. The anti-HIV drugs can be roughly grouped into two main categories (Reverse Transcriptase Inhibitors (RTI) and Protease Inhibitors (PI)). The patient is assumed to be given treatment from one of two classes of drugs, a mixture of the two treatments, or

provided no treatment. The agent starts at an unhealthy status $s_0 = [163573, 5, 11945, 46, 63919, 24]$, where the viral load and number of infected cells are much higher than the number of virus-fighting T-cells. The dynamics system is defined by a set of differential equations:

$$\dot{T}_1 = \lambda_1 - d_1 T_1 - (1 - \epsilon_1) k_1 V T_1$$

$$\dot{T}_2 = \lambda_2 - d_2 T_2 - (1 - f\epsilon_1) k_2 V T_2$$

$$\dot{T}_1^* = (1 - \epsilon_1) k_1 V T_1 - \delta T_1^* - m_1 E T_2^*$$

$$\dot{T}_2^* = (1 - \epsilon_2) N_T \delta (T_1^* + T_2^*) - cV - [(1 - \epsilon_1) \rho_1 k_1 T_1 + (1 - f\epsilon_1) \rho_2 k_2 T_2] V$$

$$\dot{E} = \lambda_E + \frac{b_E (T_1^* + T_2^*)}{(T_1^* + T_2^*) + K_b} E - \frac{d_E (T_1^* + T_2^*)}{(T_1^* + T_2^*) + K_d} E - \delta_E E$$

where $\epsilon_1 = 0.7$ (if RTI is applied, otherwise 0) and $\epsilon_2 = 0.3$ (if PI is applied, otherwise 0) are treatment specific parameters, selected by the prescribed action. See the specification of other parameters in the work of [Adams et al., 2004].

The effective period is determined by the state (mainly determined by the viral load $V$) and the treatment as follow:

$$\tau \sim \begin{cases} \text{Unif}\{7, 14\} & \text{if } V \leq 10^4 \text{ and no treatment,} \\ \text{Unif}\{3, 7\} & \text{if } V \leq 10^4 \text{ and any treatment,} \\ \text{Unif}\{3, 7\} & \text{if } 10^4 \leq V \leq 10^5 \text{ and no treatment,} \\ \text{Unif}\{3, 5\} & \text{if } 10^4 \leq V \leq 10^5 \text{ and (only RIT or only PI),} \\ 3 & \text{if } 10^4 \leq V \leq 10^5 \text{ and (both RIT and PI),} \\ 3 & \text{if } V \geq 10^5 \text{ and no treatment,} \\ \text{Unif}\{1, 2\} & \text{if } V \geq 10^5 \text{ and any treatment.} \end{cases}$$

The reward is gathered based on the patient's healthy state after effective period:

$$R(s, a, s') = -0.1V - 20000\epsilon_1^2 - 2000\epsilon_2^2 + 1000E.$$

An episode ends after $T = 1000$ days and there is no early terminated condition. The state variables are first put in log scale then normalized to have zero mean and unit standard deviation for both model training and policy optimization.

**Mujoco.** We consider the fully observable Mujoco environments, where the position of the root joint is also observed, which allows us to calculate the reward and determine the terminal condition for simulated states easily. The action repeats $\tau$ times with the following pattern:

$$\tau = \left\lfloor \frac{d - c}{2} \cos(20\pi \|\theta_v\|_2) + \frac{c + d}{2} \right\rceil$$

where $c/d$ is the minimum/maximum action repetition times, $\theta_v$ is the angle velocity vector of all joints and $\lfloor \cdot \rceil$ is rounding to the nearest integer. We have $c = 1$ for all three locomotion tasks; $d = 7$ for the swimmer and the hopper and $d = 9$ for the half-cheetah.

Because we assume the intermediate observations are not available during action repetition, the reward is calculated only based on the current observation $s$, the next observation $s'$, the control input $a$ and and repeated times $\tau$:

$$R(s, a, s', \tau) = \begin{cases} \frac{x' - x}{\tau} - 0.0001\|a\|_2^2, & \text{for the swimmer,} \\ \frac{x' - x}{\tau} - 0.001\|a\|_2^2 + \mathbb{1}(\text{alive}), & \text{for the hopper,} \\ \frac{x' - x}{\tau} - 0.1\|a\|_2^2, & \text{for the half-cheetah,} \end{cases}$$

where $x/x'$ is the previous/current position of the root joint, and there is an alive bonus of 1 for the hopper for every step. Also, instead of setting a fixed horizon, we keep the original maximum length of an episode in OpenAI Gym, i.e., the maximum number of environment steps over an episode is 1000 for all three tasks. We normalize observations so that they have zero mean and unit standard deviation for both model training and policy optimization.

# C   Experimental details

## C.1   Planning

**Learning world models.**   For all three simpler domains, we collect $N_t = 1000$ episodes as the training dataset, and collect another 100 episodes as the validation dataset. We optimize the policy for $N_e$ episodes until convergence, whose value is shown in Table 4. All final cumulative rewards are evaluated by taking the average reward of 100 trials after training policies for $N_e$ episodes.

Table 4: The choice of $N_e$ for training policies until convergence.

|             | Gridworld | Acrobot | HIV  |
|-------------|-----------|---------|------|
| model-based | 1000      | 200     | 1500 |
| model-free  | 1000      |         | 3500 |

**Model predictive control with actor-critic.**   We switch between model training and policy optimization every $M = 5000$ environment steps. Equipped with the value function from the critic, we can choose a relatively shorter planning horizon $H = 10$, which maintains the good performance while reducing the computational cost. We set a large search population $K = 1000$.

For model learning, the 90% collected trajectories are used as the training dataset and the remaining 10% are used as the validation dataset. Also, we divide a full trajectory into several pieces, whose length is equal to (or less than) the MPC planning horizon $H$. Not only does it reduce the computational cost of training a sequential model, but also helps the dynamics model provide more accurate predictions in a finite horizon.

Throughout all experiments, we use a soft-greedy trick for MPC planning to combat the model approximation errors [Hong et al., 2019]. Instead of selecting the best first action (line 20 of Algorithm 3), we take the average of first actions of the top 50 sequences as the final action for the agent. This simple approach alleviates the impact of inaccurate models and improves the performance.

**Initialization of latent states.**   While using the model for planning, initial latent states of recurrent-based models are all zeros, but they are sampled from the prior distribution (standard normal distribution) for models with the encoder-decoder structure.

## C.2   Model learning

**Scheduled sampling.**   As our model predicts the new latent state $z'$ at time $t + 1$, it needs to be conditioned on the previous state at the previous time step $t$. During training, there is a choice for the source of the next input for the model: either the ground truth (observation) or the model's own previous prediction can be taken. The former provides more signal when the model is weak, while the latter matches more accurately the conditions during inference (episode generation), when the ground truth is not known. The scheduled sampling strikes a balance between the two. Specifically, at the beginning of the training, the ground truth is offered more often, which pushes the model to deliver the accurate short term predictions; at the end of the training, the previous predicted state is more likely to be used to help the model focus on the global dynamics. In other word, the optimization objective transits from the one-step loss to multiple-step loss. Therefore, the scheduled sampling can prevent the model from drifting out of the area of its applicability due to compounding errors. We use a linear decay scheme for scheduled sampling $\epsilon = \max\{0, 1 - c \cdot k\}$, where $c = 0.0001$ is the decay rate and $k$ is the number of iterations. However, we find that the scheduled sampling only works well on the acrobot task.

**Early stopping.**   On Mujoco tasks, we utilize early stopping to prevent from overfitting, i.e., we terminate the training if the state prediction error (MSE in Equation 10) on the validation dataset does not decrease for $e$ training epochs and we use the parameters achieving the lowest state prediction error as the final model parameters. Because the model already learns the dynamics after trained for several epochs in Algorithm 3 and only needs to be refined for some novel situations, we use a linear decay scheme: $e = \max(15 - k, 3)$, where $k$ is the number of epochs in Algorithm 3. For three simpler domains, we run 12,000 iterations without early stopping.

**Model hyperparameters.** We tune the hyperparameters for dynamics models on different domains, but all baseline models use a same set of hyperparameters for comparison.

Table 5: Hyperparameters for dynamics models on different domains.

| | Gridworld | Acrobot | HIV | Swimmer | Hopper | HalfCheetah |
|---|---|---|---|---|---|---|
| Learning rate | 1e-3 | 5e-4 | 1e-3 | | | |
| Batch size | 32 | | | 128 | | |
| Latent dimension | 2 | 10 | | 128 | | 400 |
| Weight decay | 1e-3 | | | | | |
| Scheduled sampling | No | Yes | No | | | |
| GRU | one layer, unidirectional, Tanh activation | | | | | |
| Encoder hidden to latent | 5 | 20 | | | | |
| Interval timer $g_\kappa$ | N/A | | 20 | | | |
| $\lambda$ in Equation 9 | 0 | | 0.01 | | | |
| ODE network $f_\theta$ | 5 | 20 | | 128 | | 400 |
| ODE solver | Runge-Kutta 4(5) adaptive solver (dopri5) | | | | | |
| ODE error tolerance | 1e-3 (relative), 1e-4 (absolute) | | | 1e-5 (relative), 1e-6 (absolute) | | |

All hyperparameters are shown in Table 5. Specifically,

- "Learning rate" refers to the learning rate for the Adam optimizer to train the dynamics model.

- "GRU" refers to the architecture of the GRU in all experiments, including the encoder in the Latent-RNN and Latent-ODE;

- For encoder-decoder models, we use a neural network with one layer and Tanh activation to convert the final hidden state of the encoder to the mean and log variance (for applying reparameterization trick to train VAE) of the initial latent state of the decoder. "Encoder hidden to latent" refers to the number of hidden units of this neural network.

- We use a neural network with one layer and Tanh activation for the interval timer $g_\kappa$. "Interval timer $g_\kappa$" refers to the number of hidden units of this neural network.

- We use a neural network with two layers and Tanh activation for the ODE network $f_\theta$. "ODE network $f_\theta$" refers to the number of hidden units of this neural network.

- "ODE solver" refers to the numerical ODE solver we use to solve the ODE (Equations 6 and 7). Note that we do not use the adjoint method [Chen et al., 2018] for ODE solvers due to a longer computation time.

- "ODE error tolerance" refers to the error tolerance we use to solve the ODE numerically.

### C.3 Policy

**DQN hyperparameters.** The DQN for three simpler domains has two hidden layers of 256 and 512 units each with Relu activation. Parameters are trained using an Adam optimizer with a learning rate 5e-4 and a batch size of 128. We minimize the temporal difference error using the Huber loss, which is more robust to outliers when the estimated Q-values are noisy. We update the target network every 10 episodes (hard update). The action is one-hot encoded as the input for DQNs. To improve the performance of the DQN, we use a prioritized experience replay buffer [Schaul et al., 2015] with a prioritization exponent of 0.6 and an importance sampling exponent of 0.4, and its size is 1e5. To encourage exploration, we construct an $\epsilon$-greedy policy with an inverse sigmoid decay scheme from 1 to 0.05. Also, all final policies are softened with $\epsilon = 0.05$ for evaluation.

**DDPG hyperparameters.** The DDPG networks for both the actor and critic have two hidden layers of 64 units each with Relu activation. Parameters are trained using an Adam optimizer with a learning rate of 1e-4 for the actor, a learning rate of 1e-3 for the critic and a batch size of 128. The target networks are updated with the rate 1e-3. The size of the replay buffer is 1e6. To encourage exploration, we collect 15000 samples with a random policy at the beginning of training, and add a Gaussian noise $\mathcal{N}(0, 0.1^2)$ to every selected action.

Figure 5: Learning curves with all baselines on three simpler domains. The $x$-axis is the number of episodes in Algorithm 2 (for the model-free baseline, the $x$-axis is the number of episodes in the actual environment). The shaded region represents a standard deviation of average evaluation over four runs (evaluation data is collected every 4 episodes). Curves are smoothed with a 20-point window for visual clarity.

**Discount factors.** The discount factor is 0.99 for the windy gridworld and Mujoco tasks, is 0.995 for the HIV domain, and is 1 for the acrobot problem.

# D    Additional figures and tables

## D.1    Learning curves of learning world models

Figure 5 shows the learning process of all baselines on three simpler domains. Note that Figure 5 does not necessarily demonstrate the sample efficiency because the model-free method uses the online real data, whereas the model-based approach (learning world models) uses the offline real data. However, we still observe that the Latent-ODE and ODE-RNN develop a high-performing policy much faster than the model-free baseline over 1500 episodes in the HIV environment. The model-free baseline converges after 3500 episodes. In addition, the acrobot problem clearly shows the model difference in terms of abilities of modeling the continuous-time dynamics. The Latent-ODE and ODE-RNN outperforms other models;

Figure 6: Learning curves of different policies on the partially observable HIV environment.

The RNN and Latent-RNN, designed for discrete-time transitions, totally lose their ways in continuous-time dynamics, but their similar performance might suggest the limited impact of architecture (recurrent vs. encoder-decoder); The $\Delta t$-RNN and Decay-RNN also struggle on modeling continuous-time dynamics though they leverage time interval information in different ways.

Moreover, Figure 6 shows the learning process of different policies on the partially observable HIV environment. The latent policy $\pi^{MB}(a|s_{\text{partial}}, z)$ develops a better-performing policy more quickly than the vanilla model-based policy $\pi^{MB}(a|s_{\text{partial}})$ and the model-free policy $\pi^{MF}(a|s_{\text{partial}})$.

## D.2    Full results of changing time intervals

Table 6 shows the cumulative rewards of policies learned on regular measurements using pretrained models from the original irregular time schedule on three simpler domains. We can see that the Latent-ODE surpasses other baseline models in most situations.

Table 6: The cumulative reward (mean $\pm$ std, over five runs) of all baselines for all time discretizations. (a) windy gridworld; (b) acrobot; (c) HIV.

(a)

|  | RNN | $\Delta t$-RNN | Decay-RNN | Latent-RNN | ODE-RNN | Latent-ODE | Oracle |
|---|---|---|---|---|---|---|---|
| $\tau = 1$ | **-31.59 $\pm$ 1.68** | -43.21 $\pm$ 1.25 | **-31.69 $\pm$ 1.36** | -32.22 $\pm$ 1.19 | -47.31 $\pm$ 1.46 | -34.24 $\pm$ 1.93 | *-44.17 $\pm$ 0.80* |
| $\tau = 2$ | **-31.95 $\pm$ 2.10** | -39.37 $\pm$ 3.37 | -32.27 $\pm$ 2.39 | -32.75 $\pm$ 1.60 | -32.40 $\pm$ 1.43 | -33.82 $\pm$ 2.32 | *-31.61 $\pm$ 1.60* |
| $\tau = 3$ | -35.08 $\pm$ 4.91 | -34.49 $\pm$ 4.82 | -35.03 $\pm$ 4.87 | -43.74 $\pm$ 8.71 | -36.61 $\pm$ 7.04 | **-32.74 $\pm$ 2.22** | *-32.29 $\pm$ 1.60* |
| $\tau = 4$ | -38.55 $\pm$ 7.99 | -36.00 $\pm$ 5.41 | -40.58 $\pm$ 8.56 | -36.94 $\pm$ 5.73 | -64.10 $\pm$ 14.78 | **-33.26 $\pm$ 1.86** | *-32.81 $\pm$ 1.74* |
| $\tau = 5$ | -44.33 $\pm$ 9.77 | -42.97 $\pm$ 9.43 | -43.55 $\pm$ 9.50 | -36.87 $\pm$ 5.04 | -87.50 $\pm$ 18.47 | **-33.84 $\pm$ 2.02** | *-33.35 $\pm$ 2.03* |
| $\tau = 6$ | -43.14 $\pm$ 8.75 | -54.78 $\pm$ 13.35 | -53.76 $\pm$ 13.68 | -42.95 $\pm$ 9.08 | -99.36 $\pm$ 16.46 | **-35.76 $\pm$ 2.72** | *-34.17 $\pm$ 2.09* |
| $\tau = 7$ | -61.01 $\pm$ 10.03 | -64.55 $\pm$ 10.89 | -60.78 $\pm$ 10.03 | -52.32 $\pm$ 8.91 | -114.70 $\pm$ 11.65 | **-49.31 $\pm$ 6.62** | *-35.93 $\pm$ 1.95* |

(b)

|  | RNN | $\Delta t$-RNN | Decay-RNN | Latent-RNN | ODE-RNN | Latent-ODE | Oracle |
|---|---|---|---|---|---|---|---|
| $\tau = 0.2$ | -407.46 $\pm$ 13.82 | -281.92 $\pm$ 9.99 | -285.07 $\pm$ 8.47 | -237.25 $\pm$ 10.29 | -190.82 $\pm$ 9.13 | **-171.37 $\pm$ 10.07** | *-78.75 $\pm$ 3.23* |
| $\tau = 0.4$ | -268.36 $\pm$ 15.34 | -199.72 $\pm$ 12.39 | -268.33 $\pm$ 15.50 | -258.23 $\pm$ 9.60 | -128.93 $\pm$ 12.48 | **-82.24 $\pm$ 8.01** | *-48.76 $\pm$ 2.04* |
| $\tau = 0.6$ | -150.10 $\pm$ 11.59 | -168.28 $\pm$ 10.33 | -154.53 $\pm$ 11.72 | -232.42 $\pm$ 9.49 | -106.56 $\pm$ 10.74 | **-68.58 $\pm$ 8.62** | *-47.30 $\pm$ 3.95* |
| $\tau = 0.8$ | -143.98 $\pm$ 8.64 | -145.16 $\pm$ 6.57 | -180.71 $\pm$ 8.48 | -188.45 $\pm$ 8.54 | -72.00 $\pm$ 9.84 | **-63.81 $\pm$ 8.47** | *-38.18 $\pm$ 3.15* |
| $\tau = 1$ | -147.25 $\pm$ 9.55 | -66.16 $\pm$ 11.29 | -138.52 $\pm$ 13.59 | -155.81 $\pm$ 8.30 | -51.78 $\pm$ 10.16 | **-39.24 $\pm$ 7.24** | *-23.14 $\pm$ 2.50* |

(c)

|  | RNN | $\Delta t$-RNN | Decay-RNN | Latent-RNN | ODE-RNN | Latent-ODE | Oracle |
|---|---|---|---|---|---|---|---|
| $\tau = 1$ | 4.72 $\pm$ 0.55 | 2.25 $\pm$ 0.31 | 5.86 $\pm$ 0.70 | 7.02 $\pm$ 1.02 | 0.58 $\pm$ 0.07 | **7.73 $\pm$ 0.99** | *176.75 $\pm$ 56.50* |
| $\tau = 2$ | 7.28 $\pm$ 3.34 | 9.01 $\pm$ 3.29 | 8.76 $\pm$ 3.97 | 11.46 $\pm$ 5.07 | 2.80 $\pm$ 0.86 | **18.54 $\pm$ 7.10** | *70.74 $\pm$ 23.61* |
| $\tau = 3$ | 13.85 $\pm$ 4.84 | 20.10 $\pm$ 3.97 | 7.67 $\pm$ 3.47 | **26.60 $\pm$ 7.67** | 8.59 $\pm$ 2.72 | 4.95 $\pm$ 1.56 | *40.32 $\pm$ 5.56* |
| $\tau = 4$ | 7.30 $\pm$ 2.58 | **22.76 $\pm$ 4.78** | 7.20 $\pm$ 3.26 | 21.52 $\pm$ 4.59 | 12.30 $\pm$ 7.93 | **22.14 $\pm$ 2.32** | *35.58 $\pm$ 2.47* |
| $\tau = 5$ | 7.66 $\pm$ 1.79 | 17.21 $\pm$ 2.44 | 5.84 $\pm$ 1.62 | 16.95 $\pm$ 3.05 | 11.32 $\pm$ 1.09 | **21.60 $\pm$ 2.39** | *33.55 $\pm$ 1.97* |
| $\tau = 6$ | 5.36 $\pm$ 2.14 | 4.51 $\pm$ 1.41 | 3.52 $\pm$ 1.49 | 11.74 $\pm$ 3.20 | 7.82 $\pm$ 3.37 | **16.67 $\pm$ 2.79** | *19.74 $\pm$ 0.94* |
| $\tau = 7$ | 2.09 $\pm$ 0.77 | 7.34 $\pm$ 2.02 | 2.34 $\pm$ 0.97 | **9.83 $\pm$ 3.11** | 2.45 $\pm$ 0.20 | 8.21 $\pm$ 1.90 | *12.33 $\pm$ 0.91* |