[Reviews · NeurIPS 2020]

Review 1

Summary and Contributions: Authors presented two novel methods for leaning models in SMDP using neural ODE. Their method can model continuous time environment which can be further used to learn policies. Additionally, authors compared their methods with baselines and showed a significant gain in performance.

Strengths: 1. Novelty : I believe the strongest part of the paper is its novelty. Using neural ODE to learn SMDP is novel to the best of my knowledge. And tackling the SMDPs are really important as they are often the case specially in HC cases. 2. Writing : I found the paper very easy to follow and well motivated.

Weaknesses: I think the paper can significantly benefit from an experiment in a larger state space setting. As one of my main concerns is the scalability of using neuralODE for a more sophisticated environments, mainly larger state space. Transferring to environments with different time schedules : This has been mentioned, and emphasized, but I believe there should be an example of a case where this doesn't hold, they mention Atari, but I would like to see an experiment when this doesn't hold.

Correctness: Yes.

Clarity: I believe the paper is well written.

Relation to Prior Work: Yes

Reproducibility: Yes

Additional Feedback: I do wonder what is the application of this in off-policy setting, when you have irregular samples of actions. for example in HC setting and ICU, you'll probably have irregular samples of vital signs, or actions. I would appreciate a comment on that. ## After reading author's feedback I keep my score as it was.


Review 2

Summary and Contributions: The paper proposes a method for utilizing ODEs to represent dynamics for continuous-time decision-making problems with the aim of They also target filling a perceived gap in the literature of Deep RL for continuous-time problems, where most publications are model-free and discretize time if it is continuous. They claim that their approach leads to lower dependence on vast amounts of training data, better performance and that the model-based approach is well-founded. I tend to agree, although this is not exactly my area. I also believe the importance of connecting ODEs and other explicit models is critical for extending RL methods to important problems in physics, chemistry, epidemiology and population modelling.

Strengths: Their use of Neural ODEs seems novel but that is a recently introduced model so it is not surprising. The writing is very clear and the models are described in sufficient detail to be reproduced. Their experimental design and methodology is very nicely done, with increasing complexity of domains that match their claims.

Weaknesses: What they fail to seriously justify or distinguish is why the specific method of Neural ODEs are necessary. It's not even entirely convincing that their use differs from the use of ODEs to define the problem dynamics directly from the RL point of view. There are many papers on using RL to solve ODE problems, isn't any approach that attempts to learn a transition distribution where the simulator or dynamics are based upon ODEs actually a competitor here?

Correctness: Something I find unclear is the role of the latent state z and which parts of the system have access to it. It would be natural to say that the simulator is defined by an ODE, which is unknown to the agent and z is the model approximation that the agent is learning on top of their value function. But the discussion of learning the approximation model P doesn't give me confidence about that separation? does the agent ever have access to z itself or is this a true hidden state from agent's point of view? If the agent is not learning the model P then what does it matter that the ODE underlying it's experience is the "true" ODE from the simulator vs the learned neural ODE?

Clarity: The paper is very well written and each part is clearly described. There are some issues with clarity and motivation at a larger level which I have already commented on in other sections.

Relation to Prior Work: The related literature which is discussed is done well. I think an additional connection to RL approaches to solving ODE problems is very relevant.

Reproducibility: No

Additional Feedback: The rebuttal is appreciated and does clarify things somewhat. My overall review has not changed.


Review 3

Summary and Contributions: This paper proposed to model the continuous-time dynamics of the semi-Markov decision process (SMDP) via neural ordinary differential equations. Experiments across various continuous-time domains demonstrate the efficacy of the proposed methods.

Strengths: 1) The proposed method incorporate action and time into the neural ODE framework to model the continuous-time dynamics of the semi-Markov decision process (SMDP), which is theoretically sound and overcomes shortcomings of existing neural ODEs. 2) Detailed experiments on continuous-time domains demonstrates the superiority of the proposed method in terms of model learning, planning, and robustness for changes in interval times.

Weaknesses: It would be better to extend the x-axis of Fig 4 to longer environment steps to illustrate the performance more clearly. For example, the reward of the latent-ODE in the Swimmer plot seems to decrease at the end of the x-axis, while that of the delta-t RNN is shown to be increasing.

Correctness: No explicit incorrect statements.

Clarity: The paper is written clearly.

Relation to Prior Work: Yes.

Reproducibility: Yes

Additional Feedback: I have read the rebuttal and the author has addressed the issue regarding experiments.

[Author Response · NeurIPS 2020]

We thank the reviewers for thoughtful comments! We are encouraged you found our overall contribution of using neural ODEs for continuous-time RL well-motivated, novel, and well-explained. Below we respond to specific questions:

**Reviewer 1.** 1) Scalability/experiments with a larger state-space. We agree there are more interesting domains with larger state-spaces, and acknowledge that currently neural ODEs can be tough to scale. We emphasize that our main contribution is to introduce neural ODEs in a model-based RL framework for SMDPs: our experiments help us understand why neural ODEs work but other baselines fail, show benefits on standard RL benchmarks (e.g. Mujoco), and demonstrate opportunities for learning from irregular measurements and optimizing measurements. As neural ODE solvers improve in efficiency, we can expect these core contributions to carry over to larger domains.

2) Show a failure case of transferring to environments with different time schedules. Thank you for asking about limitations! We have an example in Table 6 in Supplement D.1: in some cases, (e.g. $\tau = 1$ and 2 on the HIV environment), there is a huge gap between all model-based policies and the model-free policy (oracle). This is because the model hardly sees transitions where $\tau = 1$ or 2 during training, causing poor performance when transferring to such mostly unseen time discretizations. As with any learning algorithm, one has to be careful of extrapolation. We detail additional limitations of optimizing time schedules (different from transferring to different time settings) in Supplement A.1.

3) Off-policy setting. Our training on three simpler domains (including the HIV task), learning world model schema, can be viewed as off-policy, in the sense that the model is trained on a single batch of irregularly sampled data from the environment and then agent is only trained with fictional data generated by the model and not the true environment. In the fully batch setting (learn from only one fixed set of observational data), our work can still apply but will be limited, as all batch methods are, in terms of how much it can extrapolate. However, since our training procedure for dynamics models can be used on any batch of data, we can utilize any start-of-the-art model-based off-policy RL algorithm (e.g., [1]) for planning. This is also an important point regarding the ethics and broader impact: we will emphasize this in the final paper.

**Reviewer 2.** 1) The necessity of using neural ODEs vs. regular ODEs. If we understand the question, the point is why apply RL using neural ODEs vs. ODEs in general. If we knew that a system was characterized by a particular ODE, then we could absolutely use RL to learn the parameters of that ODE. However, in many cases, e.g. healthcare contexts, we do not, and the neural ODE offers a very flexible way to define the system dynamics as an ODE-IVP problem/provide the inductive bias that the system can be modeled with an ODE without explicitly knowing its parameterization. (Apologies if we did not understand the question correctly.)

2) The role of latent variables $z$ and neural ODEs. The latent variable $z$ is from the *learned* dynamics model, *not* the environment, so it is always available to the agent. Using the learned dynamics models for planning (e.g., Dyna-style policy optimization or model predictive control), the agent can develop a good policy with less real data by relying mostly on simulating from the model; neural ODEs now let us do this for continuous-time environments.

Another note re $z$: if the state is fully observable, $z$ is just a compression of $s$. If we only receive partial observations (i.e. not Markov), $z$ will include the hidden dynamics which summarize the past (see lines 84-87 in the main paper).

**Reviewer 3.** 1) More environment steps for Swimmer. Great point! We extended Swimmer to 450k steps below. The policy developed with the Latent-ODE is still better than the one from $\Delta t$-RNN, and learns those policies faster. We could not finish extended runs for the other Mujoco in time for the rebuttal, but will do so for the final paper.

[1] Yu, Tianhe, et al. "MOPO: Model-based Offline Policy Optimization." arXiv preprint arXiv:2005.13239 (2020).

Figure 1: The learning curve for the swimmer task.

[Meta-Review · NeurIPS 2020]

The reviewers all found the paper to make a reasonable contribution. While there are some limitations that were pointed out, the rebuttal addressed them well and the experiments were appreciated.